# FAIRNESS BY DESIGN: EFFICIENT FAIR ENSEMBLES FOR LOW-DATA CLASSIFICATION

## ABSTRACT

We address the problem of fair classification in settings where data is scarce and unbalanced across demographic groups. Such low-data regimes are common in domains like medical imaging and hate speech. Our proposed method mitigates these biases by training efficient ensembles of fair classifiers on different data partitions. Aggregating predictions across ensemble members, each trained to satisfy fairness constraints, yields more consistent outcomes and stronger fairness-accuracy trade-offs than existing methods across multiple challenging medical imaging datasets, as well as on hate speech detection.

To support these findings, we provide theoretical guarantees: we prove when our fair ensembles improve performance and how much data is needed to observe these gains with statistical significance. These results extend the literature by explaining why and under what conditions ensembles improve algorithmic fairness in high-stakes applications.

## 1 INTRODUCTION

Deep learning performs exceptionally well when trained on large-scale datasets (Deng et al., 2009; Gao et al., 2020; Hendrycks et al., 2020), but its performance deteriorates in small-data regimes. This is especially problematic for marginalised groups, where labelled examples are both scarce and demographically imbalanced (D'ignazio & Klein, 2023; Larrazabal et al., 2020). In medical imaging, underrepresentation of minority groups leads to poor generalisation and higher uncertainty (Ricci Lara et al., 2023; Mehta et al., 2024; Jiménez-Sánchez et al., 2025); in hate speech detection, disparities in data availability across languages and demographics produce similar harms (Tonneau et al., 2025). As a result, the very groups most at risk of harm are those for which deep learning methods work least well.

Existing fairness interventions often fail in these low-data settings. Because data on disadvantaged groups is needed both to learn effective representations and to estimate group-specific bias, most methods underperform simple empirical risk minimisation (Zong et al., 2022).

Ensembles offer a natural way to address these challenges. By aggregating predictions across members, ensembles make more efficient use of scarce examples while leveraging disagreement between members for robustness (Theisen et al., 2023). This makes ensembles particularly attractive for fairness in low-data regimes, but without theoretical foundations, improvements remain inconsistent (Ko et al., 2023; Schweighofer et al., 2024).

We address this by introducing FAIRENSEMBLE: ensembles explicitly designed to enforce fairness constraints at the member level and provably preserve them at the ensemble level. Our theoretical results show when minimum rate and error-parity constraints are guaranteed to hold, and how much validation data is required to observe these guarantees in practice. Empirically, we demonstrate that FAIRENSEMBLE outperforms strong baselines in both medical imaging and hate speech detection—domains where fairness is urgently needed but data for disadvantaged groups is limited.

We make three contributions:

1. **Method:** We introduce an efficient ensemble framework of fair classifiers tailored to fairness in small deep learning datasets.

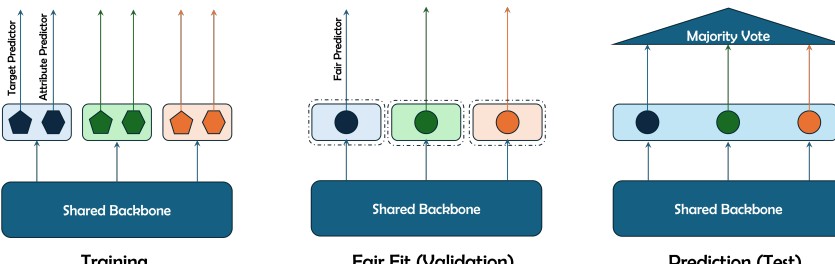

Figure 1: **FAIRENSEMBLE pipeline. Left** (*Training*): Each member shares a backbone and predicts the task label and protected attributes. **Middle** (*Validation*): We enforce a chosen fairness constraint while maximising accuracy. **Right** (*Inference*): Members vote by majority. Choices of training and validation partitions ensure that each datapoint trains some heads ensuring good generalisation. The shared backbone makes the process efficient, while Majority Voting gives theoretical guarantees.

2. **Theory:** We prove that our fair ensembles are guaranteed to preserve fairness under both error-parity and minimum rate constraints, and we derive how much data is required to observe these guarantees in practice.

3. **Results:** Across four datasets in medical imaging and hate speech detection, our method consistently outperforms existing baselines on fairness–accuracy trade-offs.

The article is organised as follows: section 2 presents related work in low-data fairness and fairness in ensembles. section 3 describes both how we construct and train the ensemble (section 3.1) and the formal guarantees for when it works (section 3.2). Finally, section 4 and section 5 provide empirical support for the benefits of fair ensembles versus strong baselines on challenging datasets.

## 2 RELATED WORK

**Fairness Challenges in Low-Data Domains**   Deep learning methods achieve near-human performance on overall metrics (Liu et al., 2020), yet consistently underperform for marginalised groups in medical imaging (Xu et al., 2024; Daneshjou et al., 2022; Seyyed-Kalantari et al., 2021) and hate speech detection (Tonneau et al., 2025). A central source of bias is unbalanced datasets (Larrazabal et al., 2020), where disadvantaged groups have too few examples to learn reliable representations, leading to poor calibration and uncertain predictions (Ricci Lara et al., 2023; Mehta et al., 2024).

Defining fairness is equally challenging. Standard parity-based metrics such as equal opportunity (Hardt et al., 2016) can be satisfied trivially by constant classifiers in imbalanced datasets and often reduce performance for all groups, a phenomenon of "levelling down" with serious real-world consequences (Zhang et al., 2022; Zietlow et al., 2022; Mittelstadt, 2019). In safety-critical domains such as medicine, *minimum rate constraints*—which enforce a performance floor across groups—are often more appropriate to ensure that classifiers serve all subpopulations (Wachter et al., 2021a). For further works, see Appendix H.

**Fairness in Ensembles:**   Prior work has observed that ensembles can sometimes improve fairness by boosting performance on disadvantaged groups (Ko et al., 2023; Schweighofer et al., 2024; Claucich et al., 2025). However, these studies are observational: improvements are not guaranteed, and in some cases ensembles can even worsen disparities (Schweighofer et al., 2024). Our approach is interventionist. Building on theoretical results for ensemble competence (Theisen et al., 2023), we extend their proofs to fairness settings. This allows us to show formally *why and when* ensembles improve fairness, rather unlike prior works which only demonstrated that they sometimes do.

## 3 METHODS

**Choice of fairness constraints.**   In this work, we focus on two fairness constraints: *equal opportunity* (the maximum difference in recall across groups; Hardt et al., 2016) and *minimum recall*

(the recall of the worst-performing group; Mittelstadt et al., 2024). Both target false negatives, which is appropriate when missing a positive case (e.g., a deadly disease) is far more costly than overdiagnosis—a scenario that is especially relevant in medical imaging (Seyyed-Kalantari et al., 2021). While we highlight these constraints, our methods and theory apply equally to other fairness metrics (see section 3.2).

## 3.1 Ensemble Construction and Training

We consider an ensemble composed of deep neural networks (DNNs) that share a pretrained convolutional backbone (Figure 1). Each ensemble member is trained on a separate fold, stratified by both the target label and group membership (T r et al., 2023). Training each member on different folds allows us to fully utilise the dataset, in contrast to standard fairness methods that require held-out validation data (Delaney et al., 2024; Buyl et al., 2023). Predictions are aggregated by majority voting, which enforces the guarantees of Theisen et al. (2023) (see section 3.2).

**Enforcing the fairness of ensemble members:** Each ensemble member is a multi-headed classifier that predicts both the task label (e.g., disease vs. no disease) and the protected attribute (i.e., group membership; see Figure 1, left). The main prediction head is trained with standard cross-entropy loss, while the auxiliary heads predict a one-hot encoding of the protected attribute using squared loss. Following the multi-head surgery of OxonFair (Delaney et al., 2024), their outputs are combined by a weighted sum. The weights are fitted on a held-out validation set to enforce fairness constraints while still maximising accuracy.

This formulation allows for any fairness group fairness definition that can be expressed of per-group confusion matrices. Because weights are selected using validation rather than training data, we can enforce error-based criteria—such as equal opportunity or minimum recall—even when the base model overfits during training.

To make fairness enforcement robust, we use a *multi-split strategy*: all non-test data is divided into a different train/validation partition per member, and fairness constraints are enforced separately on each. In practice, we optimize over accuracy together with an experiment-specific fairness constraint: either minimum recall or equal opportunity.

**Efficient ensembling of deep networks:** The main computational bottleneck in deep CNNs is the backbone. To avoid repeatedly running the same backbone for different ensemble members, we concatenate all classifier heads on top of a shared backbone. During training, the loss is masked so that only the relevant head is updated for each data point. When the backbone is pretrained and frozen,[1] this procedure is effectively equivalent to training each ensemble member independently, while requiring only a single backbone pass. A related idea with backbone fine-tuning is described by Chen & Shrivastava (2020). We use EfficientNetV2 (Tan & Le, 2021) pretrained on ImageNet (Deng et al., 2009) as the backbone in all experiments.

This design yields substantial efficiency gains. Inference speed is essentially identical to a single ERM model, while training is somewhat slower due to multiple heads, but still much faster than training all members separately (which would be about $M \times$ slower for an M-member ensemble). Appendix F provides empirical comparisons, and Appendix A gives implementation details. To ensure robustness, each experiment is repeated over three train/test splits.

## 3.2 Formal Guarantees for Fairness

We now ask: under what conditions can ensembles be expected to *guarantee* fairness improvements? As mentioned in section 2, prior work on fairness in ensembles is observational, showing that ensembles sometimes improve fairness (Claucich et al., 2025; Ko et al., 2023, e.g.,). In contrast, we provide theoretical conditions under which fairness is improved, together with guidance on how these guarantees can be implemented in practice.

Specifically, we address two questions:

---

[1]Freezing the backbone helps prevent overfitting on small datasets.

1. **Minimum rate constraints:** How high must minimum rate constraints be set to ensure that ensembles preserve fairness?

2. **Sample sizes:** How large must group sizes in the validation and evaluation sets be to observe these guarantees empirically?

The core theory is based on the work of Theisen et al. (2023), who show that *competent* ensembles never hurt performance. Inofrmally, an ensemble is competent if it is more likely to be confidently right than confidently wrong. Formally, let the error rate of an ensemble $\rho$ be:[2]

$$W_\rho = W_\rho(X, Y) = \mathbb{E}_{h\sim\rho}[1(h(X) \neq Y)]$$

and define the competence of $\rho$ as:

$$C_\rho = P(W_\rho \in [t, 1/2)) - P(W_\rho \in [1/2, 1-t])$$

We say that the ensemble is *competent* if $C_\rho \geq 0$. This definition makes no distributional assumptions and can be verified on a held-out evaluation set. Theisen et al. (2023) show that if competence holds on a dataset $(X, Y)$, then majority voting improves performance relative to a single classifier, with the improvement bounded by the disagreement between ensemble members.

To extend competence to fairness metrics, we evaluate competence on *restricted subsets of the data*. Let $\mathcal{G}$ be the set of protected groups. For any group $g \in \mathcal{G}$, write $g+$ for the positives ($Y = 1, A = g$) and $g-$ for the negatives ($Y = 0, A = g$). We then define the restricted ensemble error

$$W_\rho^{g+} = \mathbb{E}_{h\sim\rho,(X,Y)\sim\mathcal{D}_{g,+}}[\,\mathbf{1}\{h(X) \neq Y\}\,]$$

and say the ensemble is *restricted groupwise competent* if

$$C_\rho^{g+} = P(W_\rho^{g+} \in [t, 1/2)) - P(W_\rho^{g+} \in [1/2, 1-t]) \geq 0 \ \ \forall g \in \mathcal{G} \tag{1}$$

Minimum recall corresponds to competence on $g+$, minimum sensitivity to competence on $g-$, and overall accuracy to competence on the full dataset.

Based on this, we derive three main results:

1. **Minimum rate constraints:** If an ensemble is restricted groupwise competent, and every member of the ensemble satisfies a minimum rate constraint, then the ensemble as a whole also satisfies that minimum rate.

2. **Error parity:** If an ensemble is restricted groupwise competent, and if every member of the ensemble approximately satisfies an error parity measure (e.g., equal opportunity), then the ensemble as a whole also approximately satisfies it. The achievable bounds depend on disagreement- and error rates of the members.

3. **Independent errors:** If an ensemble is *not* restricted groupwise competent, but the errors made by the ensemble are independent, enforcing a minimum recall rate of $k \geq 50\%$ on every member of the ensemble guarantees that the ensemble also has a minimum recall rate of $k$.

Together, these results show how ensemble competence on restricted subsets provides guarantees for both minimum rate constraints and error parity measures, covering a broad range of fairness definitions.

### 3.2.1 RESTRICTED GROUPWISE COMPETENCE GUARANTEES

**1. Minimum rates for competent ensembles:** The proofs of Theisen et al. (2023) are dataset-agnostic: if an ensemble is competent on any dataset, then ensembling on that dataset does not

---

[2]For definitions of all notation used see Table 5.

decrease the average accuracy. Applying the definition of competence to the restricted subset $g+$, accuracy on that subset corresponds exactly to recall.

The core theorem from Theisen et al. (2023) bounds the *Error Improvement Rate (EIR)*—the ensemble's relative improvement over a single classifier—by the *Disagreement Error Ratio (DER)*. See Appendix C for formal definitions. For binary classification, the bounds are given by Eq. 2 for an arbitrary data distribution, $\mathcal{D}$:

$$\text{DER}_{\mathcal{D}} \geq \text{EIR}_{\mathcal{D}} \geq \max(\text{DER}_{\mathcal{D}} - 1, 0) \tag{2}$$

Since there are no assumptions about the distribution of $\mathcal{D}$, we can restrict it to $g+$. Since the EIR is always non-negative, it follows that the minimum recall of a competent ensemble is at least as big as its members.

**2. Error parity from competence:** Error-parity constraints such as approximate equal opportunity (equality of recall across groups; Hardt et al., 2016) or approximate equality of accuracy (Zafar et al., 2019) are harder to guarantee. The difficulty is that while ensembles can improve average performance, unequal improvements across groups can increase disparities (see, e.g., Schweighofer et al., 2024). Nonetheless, competence still yields limited but useful bounds.

We consider the $L_{\infty}$ form of approximate fairness: a classifier has $k$-approximate fairness with respect to groups $\mathcal{G}$ if
$$k \geq \max_{g \in \mathcal{G}} L_g(h) - \min_{g \in \mathcal{G}} L_g(h) \tag{3}$$
where $L_g$ is the average loss on group $g$, corresponding to 1 minus one of the measures we are concerned with (typically recall).

The question then is, if every member of the ensemble exhibits $k$-approximate fairness, what fairness bounds do we have for the ensemble? By applying Eq. 2 (see Appendix G.2 for derivation), we obtain the following bound:

$$k^* \leq k + \max_{g \in \mathcal{G}} \mathbb{E}_{h \sim \rho}[L_g(h)]\text{DER}_{g^*} - \max(0, \min_{g \in \mathcal{G}} \mathbb{E}_{h \sim \rho}[L_g(h)](\text{DER}_{g^*} - 1)) \tag{4}$$

Both bounds are pessimistic. In practice, our approach works well for enforcing equal opportunity (see 5). Still, two insights follow: First, the same bounds apply to any group-based error-parity measure (not just equal opportunity). Second, because bound scales with $L_g$, the worst-case disparity shrinks as group losses decrease. In practice, this means that enforcing sufficient minimum rate constraints through our method can tighten the bounds.

### 3.2.2 GUARANTEES FOR MINIMUM RECALL

A key challenge is that while members of an ensemble are often competent on the full dataset, the proportion of positively labelled data is small in many critical settings (such as medical imaging). When restricting to the subset $g+$ (positives in group $g$), competence may thus fail to hold.

In such cases, we can restore competence by enforcing sufficiently high minimum recall rates. Recall is simply accuracy restricted to positives, and raising this rate ensures that the conditions of *Jury Theorems* apply. These results generalise the classic Condorcet Jury Theorem (du texte Condorcet, 1785), which shows that majority vote of independent voters who are each more likely to be right than wrong (i.e., accuracy $> 0.5$) improves over the average voter, with the accuracy converging to 1 as the number of voters increases. Modern variants extend this to heterogeneous accuracies and mildly correlated voters. (Berend & Paroush, 1998; Kanazawa, 1998; Pivato, 2017).

For completeness, we sketch the proof in the simple case of independent classifiers with mean recall above 0.5. Let $K$ be the number of positive predictions in an ensemble with $N$ members for a given data point $x$. We model each classifier prediction as $K_i \sim \text{Bernoulli}(p_i)$, where $p_i = k + \delta$ and $\delta \geq 0$ reflects the enforced minimum recall margin. The mean recall is then

$$\bar{p} = \frac{1}{N} \sum_{i=1}^{N} p_i.$$

**Lemma 1** (Ensemble competence under minimum rate constraints). *If $N$ is odd and $\bar{p} \geq 0.5$, then*

$$P(K > N/2) \ \geq \ P(K < N/2).$$

*Proof.* If $\bar{p} = 0.5$, then $\mathbb{E}[K] = N/2$ and the distribution is symmetric. Since $N$ is odd, $P(K = N/2) = 0$ and hence $P(K > N/2) = P(K < N/2) = 1/2$.

For $\bar{p} > 0.5$, define

$$F(p_1, \ldots, p_N) = P\left(\sum_{i=1}^{N} K_i > N/2\right).$$

This function is monotone non-decreasing in each $p_i$. If at least one $p_i > 0.5$, then $F$ strictly exceeds $1/2$, implying

$$P(K > N/2) > 1/2 > P(K < N/2).$$

$\square$

This shows that enforcing minimum recall above $0.5$ guarantees ensemble competence on the positives. More generally, by setting sufficiently high minimum rate constraints (see section 3.1), our ensembles preserve fairness by construction–providing formal guarantees rather than the empirical observations of Ko et al. (2023); Schweighofer et al. (2024).

**3. Minimum recall under Independent Errors:** The lemma above shows that an ensemble is competent whenever its members have a mean recall above $0.5$. Under the additional assumption of independent errors, this implies that enforcing a minimum recall rate $k > 0.5$ for each group and each member is sufficient to guarantee that the ensemble also achieves recall of at least $k$. Our multi-split enforcement strategy (Sec. 3.1) ensures exactly this: every classifier head is tuned on validation data to meet the required minimum recall, so that majority voting preserves the guarantee at the ensemble level.

### 3.2.3 MINIMUM VALIDATION AND EVALUATION SIZES

Under the assumption of independent errors, a minimum recall of $k > 0.5$ on the test set, guarentees that the ensemble will also have a minimum recall of $k$. The challenge here is that recall constraints are imposed on validation data, and as we are dealing with very low-data groups, sometimes with $< 100$ positive cases, the constraints need not generalise to test data.

To ensure these constraints generalise to test data, we want to determine the minimum recall, $P_{\min}$, required the on a validation set with $m$ positives in the minority group such that with a probability $\alpha$, the recall on an evaluation set with $n$ positives will be at least $50\%$. This will guarantee that the minimum recall of the ensemble is greater than the average recall of each member. We assume that validation and test sets are of known sizes, $m$ and $n$ respectively, and drawn from the same distribution. By drawing on the literature for one-sided hypothesis tests on Bernoulli distributions, we arrive at Eq. 5.

$$p_{\min} = 0.5 + \tfrac{1}{2} z_{1-\alpha} \sqrt{\tfrac{1}{m} + \tfrac{1}{n}}. \tag{5}$$

Where $z_{1-\alpha}$ is the z-score for significance $1-\alpha$. The primary implication of Eq. 5 is that to maximise the size of the training set, one should set $m \approx n$ – especially for small data. For derivations see Appendix G.1. We find empirical support for our theoretical guarantees of fairness on positive samples in Appendix E. Here, we show that as long as the minimum recall is enforced at a sufficiently high threshold, we observe restricted groupwise competence on the test set.

## 4 EXPERIMENTAL SETUP

### 4.1 DATA AND PROTECTED ATTRIBUTES

We evaluate on three medical imaging datasets from MedFair (Zong et al., 2022) and FairMedFM (Jin et al., 2024). Each task is a binary classification with image-only inputs (discarding any auxiliary

Table 1: Evaluation datasets. "Min. Positives" is the number of *positive* examples in the smallest group (bold). These small counts stress-test low-data fairness.

| Dataset | Task | # Min. Positives | Protected Attributes |
|---|---|---|---|
| *Medical Imaging* | | | |
| HAM10000 | Skin cancer | 94 | Age (0-40, **40-60**, 60+) |
| Fitzpatrick17k | Dermatology | 60 | Skin type (I-IV, V, **VI**) |
| Harvard-FairVLMed | Glaucoma | 399 | Race (Asian, White, **Black**) |
| *Natural Language* | | | |
| Multilingual Twitter | Polish hate speech | 60 | Gender (male, **female**) |

features for fair comparison). We add a multilingual hate speech dataset for cross-modality validation (Huang et al., 2020).

For Fitzpatrick17k, the common binary split (I–III vs. IV–VI) can mask harms to the darkest tone (VI), which comprises only 0.4% of positives. We therefore separate V and VI, grouping I–IV to preserve adequate support elsewhere.

**Preprocessing and splits:** Images are center cropped and resized to 224x224 (Deng et al., 2009) with random augmentations during training. Dataset-specific validation/test sizes follow section 3.2.3 to guarantee 70% minimum observable recall. See Appendix A for full details.

**Hate speech:** We use the Multilingual Twitter Corpus (Huang et al., 2020) and Delaney et al. (2024): On Polish,we enforce 5% equal opportunity on Polish data using perceived gender as the protected attribute. This helps show the generality of our method.

## 4.2 EVALUATION METRICS

Medical classification is a non-zero-sum game where "levelling down"—reducing all groups' performance to achieve parity—can have fatal consequences (Mittelstadt et al., 2024). The crucial harm is failing to diagnose ill people from disadvantaged groups, making minimum *recall* the appropriate metric rather than disparity-based measures like equal opportunity. Moreover, with positive class incidence below 10% for disadvantaged groups, a trivial all-negative classifier would achieve high accuracy, and perfectly satisfy equal opportunity, while missing all sick patients.

We evaluate models on the Pareto frontier between minimum recall and accuracy (Delaney et al., 2024). Our primary metric, $\mathrm{FairAUC}$, summarizes this frontier by computing the best accuracy $a$ achievable at each minimum recall threshold $t \in T$:

$$\mathrm{FairAUC} = \frac{1}{|T|} \sum_{t \in T} \left( \max_{(a,r) \in M, r \geq t} a \right) \tag{6}$$

where $M$ are model configurations and $r$ is minimum recall. We evaluate over $T \in [0.5, 1]$—the zone with theoretical guarantees (section 3.2). Confidence intervals are computed via 200 bootstrap samples at 95% level. For baselines without explicit thresholding, we generate Pareto frontiers by varying prediction thresholds on validation data. $\mathrm{FairAUC}$ is not defined for error-parity measures.

## 4.3 BASELINES AND ENSEMBLE SETTINGS

We compare against a set of established fairness methods to ensure a meaningful contribution. As a reference **Empirical Risk Minimisation (ERM)** simply minimises training error without considering fairness (Vapnik, 2000). We further include **Domain-Independent Learning**, which trains a separate classifier for each protected class with a shared backbone, and **Domain-Discriminative Learning**, which encodes protected attributes during training and removes them at inference (Wang et al., 2020). **Fairret** introduces a regularisation term that accounts for the protected attribute and fairness criterion (Buyl et al., 2023), while **OxonFair** tunes decision thresholds on validation data to enforce group-level fairness (Delaney et al., 2024). Finally, we include an **ERM Ensemble**, which is equivalent to our method without attribute predictors to enforce fairness.

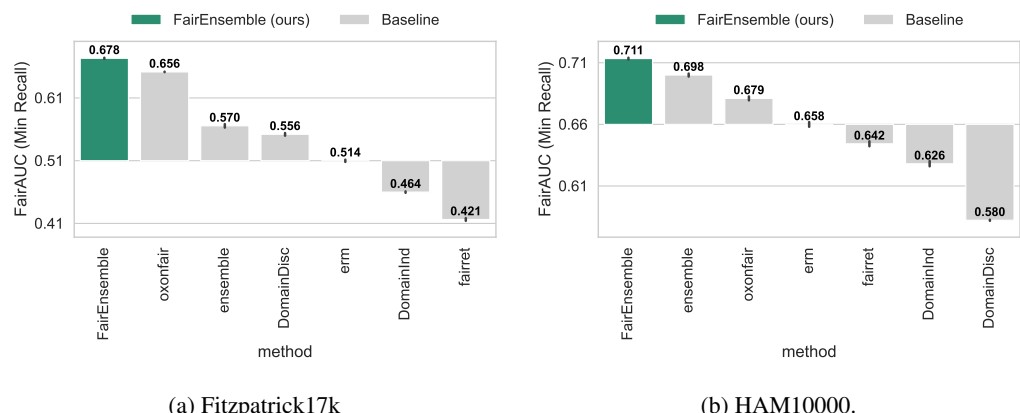

(a) Fitzpatrick17k             (b) HAM10000.

Figure 2: **Fairness–accuracy AUC (FairAUC) relative to ERM.** FAIRENSEMBLE achieves higher FairAUC than all baselines on Fitzpatrick17k (left) and HAM10000 (right). Error bars show 95% bootstrap CIs. Evaluation follows section 4.2 over minimum-recall thresholds in $[0.5, 1]$.

All baselines are trained with the same configuration as our ensembles. Minority groups are rebalanced through upsampling, and we reimplement methods following Zong et al. (2022) and Delaney et al. (2024). For Fairret, we perform a hyperparameter search over regularisation weights. To generate comparable Pareto frontiers, we fit global prediction thresholds so that a minimum recall of $k$ is enforced on a held-out validation set, mirroring deployment where thresholds are tuned on available data but applied to unseen test data (Kamiran et al., 2013). For hate speech, we compare directly against baselines reported by Delaney et al. (2024).

**Ensemble size:** We use 21 members for all ensembles. Appendix D shows that $\mathrm{FairAUC}$ is stable across different sizes from 3 to 21 within confidence intervals. We therefore default to the larger size: it is consistent with our theory that majority voting benefits from more members, while our shared-backbone design keeps inference time essentially unchanged (see Appendix F).

## 5  RESULTS

### 5.1  MEDICAL IMAGING

Table 2: Accuracy and fairness violations. Best value in **bold**.

| Dataset | Accuracy ↑ | | Fairness Violations ↓ | |
|---|---|---|---|---|
| | FAIRENSEMBLE | OxonFair | FAIRENSEMBLE | OxonFair |
| Fairvlmed | **0.665** | 0.657 | **0.009** | 0.011 |
| Fitzpatrick17K | **0.642** | 0.623 | 0.057 | **0.048** |
| Ham10000 | **0.707** | 0.679 | **0.067** | 0.082 |

**FairVLMed:** In Figure 3 (right), only our FAIRENSEMBLE method maintains fairness at strict thresholds (EqualOpportunity $< 4\%$). Most other methdos break down above 6%. Compared to OxonFair, FairEnsemble keeps higher accuracy with lower fairness violations (Table 2). While standard ensembles have slightly higher accuracy, our fair ensembles consistently reduce disparities further (e.g., equal opportunity from 6% to $< 5\%$ with $< 1\mathrm{pp}$ accuracy loss).

**Fitzpatrick17k:** For Fitzpatrick17k, where there are only 60 positive samples from the darkest skin type (VI), FAIRENSEMBLE clearly outperforms all baselines. Our best variant reaches $\mathrm{FairAUC} = 67.7\%$, compared to 57.0% for standard ensembles and 51.3% for ERM (Figure 2a. Across thresholds, FAIRENSEMBLE is consistently Pareto-optimal (Figure 3, centre).

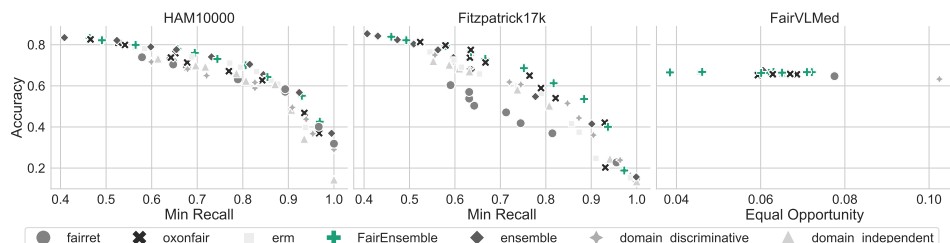

Figure 3: **Pareto frontiers across datasets.** FAIRENSEMBLE (green) yields more stable fairness–accuracy trade-offs than baselines (grey). Left/centre: minimum recall (HAM10000, Fitzpatrick17k). Right: equal opportunity (FairVLMed). See section 4.2 for metric definition.

**HAM10000:** Table 2 shows FAIRENSEMBLE achieves both the highest accuracy and lowest fairness violations on HAM10000. Its FairAUC = 71.1% significantly outperforms ERM (65.7%), standard ensembles (69.7%), and OxonFair (67.9%). All other baselines perform worse than ERM.

## 5.2 NLP: HATE SPEECH DETECTION

Table 3: Comparison against baselines from Delaney et al. (2024)

|  | Base | CDA | DP | EO | Dropout | Rebalance | OxonFair | Ensemble | FAIRENSEMBLE |
|---|---|---|---|---|---|---|---|---|---|
| Acc. (↑) | **89.80** | **89.80** | 89.50 | 89.10 | 88.90 | 89.50 | 88.50 | 89.70 | 87.76 |
| DEO (↓) | 21.40 | 16.00 | 17.90 | 13.20 | 13.80 | 19.10 | 8.45 | 17.17 | **5.68** |

The results for hate speech detection are shown in Table 3. We compare against the baselines reported by Delaney et al. (2024) on Polish data, where the task is to detect hate speech with perceived gender as the protected attribute. The fairness constraint is equal opportunity, measured by the difference in equal opportunity (DEO), with a target of DEO < 0.05.

Two main findings stand out. First, our FAIRENSEMBLE achieves the lowest disparity (DEO = 5.68%), comfortably satisfying the fairness constraint while incurring only a modest 1.5% drop in accuracy compared to the strongest baselines. OxonFair, optimised on the same constraint, suffers larger violations. Second, a standard **Ensemble** without fairness surgery slightly improves accuracy over ERM, but fails to reduce disparity (DEO = 17.17%).

In short, we reliably enforce fairness in NLP tasks: it substantially improves fairness where ensembles alone do not, demonstrating that our guarantees extend beyond medical imaging to text classification.

## 6 CONCLUSION

We have presented a novel framework for constructing efficient ensembles of fair classifiers that address the challenge of enforcing fairness in low-data settings. Across three medical imaging datasets and a multilingual hate speech dataset, our method consistently outperforms existing fairness interventions on fairness-accuracy trade-offs. Unlike prior work on ensembles that observed occasional fairness improvements, our approach guarantees that fairness is not degraded and shows that ensembles are a practical tool for reusing scarce data to produce more reliable fairness estimates.

Our theoretical analysis explains *why* these improvements occur. We prove that enforcing minimum rate constraints above 0.5 ensures ensemble competence for the worst-performing groups, derive bounds for error-parity measures such as equal opportunity, and provide principled guidance on the validation and test sizes needed for these guarantees to hold in practice. Together, these results expand the understanding of both when and why ensembles improve fairness, offering a principled and empirically validated method for building more equitable classifiers in high-stakes domains.

ETHICS STATEMENT

While we show substantial fairness improvements on the benchmarks, this does not necessarily translate into clinical improvements (Allen et al., 2019; Chien et al., 2022). Fair models are only one piece in a complex socio-technical pipeline for ensuring just deployment (Selbst et al., 2019). Nevertheless, we aim to provide the best grounds for such clinical studies through reproducible pipelines and well-documented assumptions.

Furthermore, like other fairness methods, we are limited by the need to access protected attributes during training. For many scenarios, access to this data might not be possible, and annotations or proxies thereof can often be noisy.

Our proposed fair ensemble method pertains only to the algorithmic fairness of the classifiers. It is important not to use it as a techno-fix to mask underlying disparities (Selbst et al., 2019). Instead, it should be seen as a tool that can help ensure that socio-technical classification systems adhere to the fairness requirements relevant within that context (Wachter et al., 2021a;b).

REPRODUCIBILITY STATEMENT

We have gone to great lengths to ensure the reproducibility of all results from the paper. Full (anonymised) source code with detailed instructions for running the code can be found at `https://anonymous.4open.science/r/guaranteed-fair-ensemble-82B1/README.md`. Implementation details are in Appendix A. Information on Data Access is in Appendix B. Complete definitions of formalisms used are in Appendix C.

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

## A    IMPLEMENTATION DETAILS

The code and instructions for reproducing the results can be found in an anonymised GitHub repository[3]. Optimisation for all models is done using Adam (Kingma & Ba, 2015) with a learning rate of 0.0001.

The test splits for the baseline methods (see section 4.3 were all with the same seed as the first run of the ensembles. All experiments where run with deterministic seeds for reproducibility (see repository).

---

[3]Link: anonymous.4open.science/r/guaranteed-fair-ensemble-82B1

To choose the sizes of the validation and test sets, we use the theory described in section 3.2.3. Applying a minimum observable recall of 70%, we get the below sizes. These were applied consistently across all methods.

- **Fitzpatrick17K**: $|\mathcal{D}_{\text{valid}}| = 33\%, |\mathcal{D}_{\text{test}}| = 25\%$
- **HAM10000**: $|\mathcal{D}_{\text{valid}}| = 20\%, |\mathcal{D}_{\text{test}}| = 20\%$
- **FairVLMed**: $|\mathcal{D}_{\text{valid}}| = 10\%, |\mathcal{D}_{\text{test}}| = 10\%$

For fairret, we do evaluate over a set of regularisation parameters ranging, which include [0.5, 0.75, 1.0, 1.25, 1.5]. While Buyl et al. (2023) technically doesn't require a validation set, it makes use a hyperparameter to govern the fairness/accuracy trade-off. This hyperparameter can not be set a priori, and must be tuned for every dataset, requiring the use of validation data. We do no additional parameter search for Domain Discriminative, ERM, or Domain Independent.

All training was done on a single H100. For the final results of the paper, we ran analysis on 3 datasets for 3 iterations. Using Weights & Biases (Biewald, 2020), we can see that each run took ĩ1 minutes. In addition, the baseline experiments add an extra 20 runs. In total this results in approximately 14.5 hours of compute to reproduce the complete results. Note, that the experiments could easily have been run on cheaper hardware since the EfficientNetV2 models only have 43M parameters.

While the above details the compute used to produce the results from the paper, further experiments were made prior to this. Particularly, we experimented with a less efficient ensemble structure requiring a separate run for each ensemble member. This required significantly more compute time.

## B  DATA ACCESS AND INFORMATION

We provide links for accessing the data in Table 4. Note, that while all data is openly available for academic research, some of it requires approval by the providers.

For detailed summary statistics for HAM10000 and Fitzpatrick17k, we refer to the supplemental material in MedFair (Zong et al., 2022). For FairVLMed, we refer to the FairCLIP paper (Luo et al., 2024) as well as the GitHub page. For further details, see the original publications.

Table 4: Dataset access information

| Dataset | Access URL | Reference |
|---|---|---|
| Fitzpatrick17k | `https://github.com/ mattgroh/fitzpatrick17k` | (Groh et al., 2021) |
| HAM10000 | `https://dataverse. harvard.edu/dataset. xhtml?persistentId=doi: 10.7910/DVN/DBW86T` | (Tschandl et al., 2018) |
| FairVLMed | `https://github.com/ Harvard-Ophthalmology-AI-Lab/ FairCLIP` | (Luo et al., 2024) |

## C  THEORETICAL FORMALISMS

Table 5 defines all notation used in the main paper.

As mentioned in the main paper, Theisen et al. (2023) bound the improvements of an ensemble (i.e., the *Ensemble Improvement Ratio (EIR)*) by the *Disagreement-Error Ratio (DER)* of the ensemble, i.e., the ratio of the average pairwise disagreement rate to the average error of ensemble members.

For completeness, we repeat their major results below. Note that while Theisen et al. (2023) considers a fixed distribution $\mathcal{D} = (X, Y)$, which they frequently drop from their notation, we preserve it as we will want to vary $\mathcal{D}$.

Table 5: Summary of notation used in section 3.2.

| Symbol | Definition |
|---|---|
| $\mathcal{D}$ | Data distribution over $(X, Y)$ |
| $X$ | Input features |
| $Y \in \{0, 1\}$ | Binary label (1 = positive, 0 = negative) |
| $A \in \mathcal{G}$ | Protected attribute; $\mathcal{G}$ is the set of groups |
| $g \in \mathcal{G}$ | A particular protected group |
| $\mathcal{D}_{g,+}, \mathcal{D}_{g,-}$ | Conditional distributions $\mathcal{D}\,|\,(A = g, Y = 1)$ and $\mathcal{D}\,|\,(A = g, Y = 0)$ |
| $g+, g-$ | Shorthand for positives $(A = g, Y = 1)$ and negatives $(A = g, Y = 0)$ |
| $h$ | Individual classifier (ensemble member) |
| $h'$ | Another (distinct) ensemble member |
| $\rho$ | Distribution over ensemble members (uniform in practice) |
| $h_{\mathrm{MV}}$ | Majority-vote classifier induced by $\rho$ |
| $N$ | Ensemble size (number of members) |
| $L_{\mathcal{D}}(h)$ | Error rate (0–1 loss) of $h$ on $\mathcal{D}$ |
| $L_g(h)$ | Groupwise loss on group $g$ (e.g., $1 -$ recall or $1 -$ accuracy) |
| $D_{\mathcal{D}}(h, h')$ | Disagreement rate between $h$ and $h'$ on $\mathcal{D}$ |
| $W_\rho(X, Y)$ | Ensemble error rate on $\mathcal{D}$: $\mathbb{E}_{h\sim\rho}[\mathbf{1}\{h(X) \neq Y\}]$ |
| $W_\rho^{g+}$ | Ensemble error rate on positives in group $g$ (i.e., on $\mathcal{D}_{g,+}$) |
| $W_\rho^{g-}$ | Ensemble error rate on negatives in group $g$ (i.e., on $\mathcal{D}_{g,-}$) |
| $t \in [0, 1/2]$ | Margin parameter in competence definitions |
| $C_\rho$ | Competence on $\mathcal{D}$: $P(W_\rho \in [t, 1/2)) - P(W_\rho \in [1/2, 1-t])$ |
| $C_\rho^{g+}$ | Restricted groupwise competence on $g+$ (analogously $C_\rho^{g-}$ for $g-$) |
| $\mathrm{EIR}_{\mathcal{D}}$ | Error Improvement Rate: $\frac{\mathbb{E}_{h\sim\rho}[L_{\mathcal{D}}(h)] - L_{\mathcal{D}}(h_{\mathrm{MV}})}{\mathbb{E}_{h\sim\rho}[L_{\mathcal{D}}(h)]}$ |
| $\mathrm{DER}_{\mathcal{D}}$ | Disagreement–Error Ratio: $\frac{\mathbb{E}_{h,h'\sim\rho}[D_{\mathcal{D}}(h,h')]}{\mathbb{E}_{h\sim\rho}[L_{\mathcal{D}}(h)]}$ |
| $g^*$ | Index for the distribution on which DER/EIR are computed (e.g., $g+$, $g-$, or full) |
| $k$ | Minimum rate constraint (e.g., minimum recall/sensitivity) |
| $k^*$ | Upper bound on ensemble fairness gap under error-parity bounds |
| $K$ | Number of positive predictions among $N$ members for a datapoint |
| $K_i$ | Bernoulli indicator of the $i$-th member's positive prediction |
| $p_i$ | Success prob. of $K_i$; $p_i = k + \delta$ under enforced minimum rate |
| $\bar{p}$ | Mean recall across members: $\bar{p} = \frac{1}{N}\sum_{i=1}^{N} p_i$ |
| $\delta \geq 0$ | Margin by which enforced minimum rate exceeds $k$ on validation |
| $m, n$ | # positives in validation/test for the minority group (for power analysis) |
| $\alpha$ | Significance level in the one-sided test |
| $z_{1-\alpha}$ | $(1 - \alpha)$-quantile of the standard normal distribution |
| $p_{\min}$ | Minimum observed validation recall to ensure test-time recall $> 0.5$: $p_{\min} = 0.5 + \frac{1}{2}z_{1-\alpha}\sqrt{\frac{1}{m} + \frac{1}{n}}$ |

Their results are as follows:

The ensemble improvement rate is defined as:

$$\mathrm{EIR}_{\mathcal{D}} = \frac{\mathbb{E}_{h\sim\rho}[L_{\mathcal{D}}(h)] - L_{\mathcal{D}}(h_{\mathrm{MV}})}{\mathbb{E}_{h\sim\rho}[L_{\mathcal{D}}(h)]}. \tag{7}$$

and the Disagreement-Error Ratio as:

$$\mathrm{DER}_{\mathcal{D}} = \frac{\mathbb{E}_{h,h'\sim\rho}[D_{\mathcal{D}}(h, h')]}{\mathbb{E}_{h\sim\rho}[L_{\mathcal{D}}(h)]}. \tag{8}$$

Where $L_{\mathcal{D}}(h)$ is the error rate for classifier, $h$, on data distribution, $\mathcal{D}$, $h_{\mathrm{MV}}$ is the majority vote classifier, $\mathbb{E}_{h\sim\rho}$ indicates the expected value over all ensemble members, and $\mathrm{D}_{\mathcal{D}}(h, h')$ is the disagreement rate between classifiers, $h$ and $h'$.

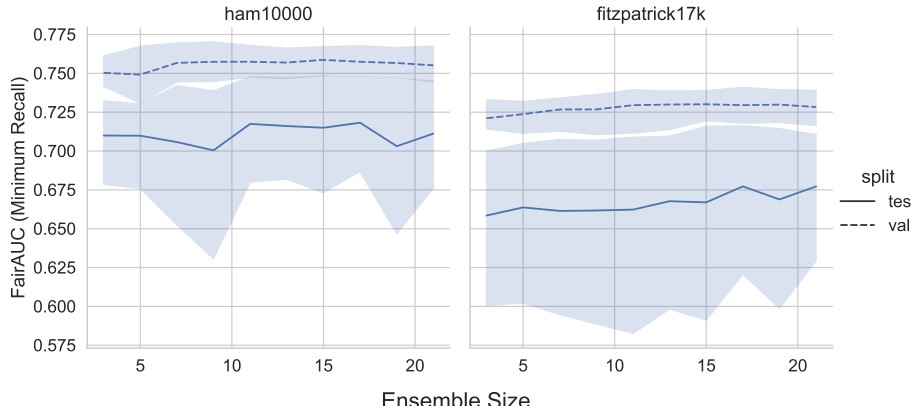

Figure 4: Relationship between **Ensemble Size** (X-axis) and **FairAUC** (Y-axis) across two datasets. No significant relationship is observed.

Specifically, the authors provide upper and lower bounds on the EIR. Crucially, this rests on an assumption of *competence*, which informally states that ensembles should always be at least as good as the average member. More formally, Theisen et al. (2023) state:

**Assumption 1** (Competence). *Let $W_{\rho,\mathcal{D}} \equiv W_\rho(X,Y) = \mathbb{E}_{h \sim \rho, \mathcal{D}}[\mathbf{1}(h(X) \neq Y)]$. The ensemble $\rho$ is* competent *if for every $0 \leq t \leq 1/2$,*

$$\mathbb{P}(W_{\rho,\mathcal{D}} \in [t, 1/2)) \geq \mathbb{P}(W_{\rho,\mathcal{D}} \in [1/2, 1-t]). \tag{9}$$

This assumption can be interpreted as formalising the statement that a majority voting ensemble is more likely to be confidently right than confidently wrong.

Based on this assumption, Theisen et al. (2023) prove the following theorem:

**Theorem 1.** *Competent ensembles never hurt performance, i.e., EIR $\geq 0$.*

This assumption is only required to rule out pathological cases. For most real-world examples, this will be trivially satisfied. In the case of binary classification, the bounds on EIR can be simplified to Eq. 2 from the main text.

## D    ABLATION: ENSEMBLE SIZES

In this section, we ask: "How does ensemble size affect performance?" We examine how $\mathrm{FairAUC}$ varies with ensemble size on the test set, and whether validation performance predicts test performance.

Our design makes this straightforward: because ensemble members are trained independently, we can form smaller ensembles by subsampling members. We construct ensembles of size $m \in \{3, 5, \ldots, M\}$ with $M = 21$, and compute $\mathrm{FairAUC}$ on both validation and test sets for HAM10000 (Tschandl et al., 2018) and Fitzpatrick17k (Groh et al., 2021) across all train/test partitions.

Figure 4 shows no consistent trend: confidence intervals are wide, and performance does not vary systematically with ensemble size. An alternative heuristic is to use validation $\mathrm{FairAUC}$ to select ensemble size, but as Figure 5 shows, the relationship between validation and test performance is too noisy to be useful. This is expected, as our method already leverages all non-test data to fit fairness weights.

Lacking a strong empirical heuristic, we adopt the largest ensemble ($M = 21$), which best aligns with our theoretical results: larger ensembles provide stronger guarantees under Jury-theorem arguments (see section 3.2.2).

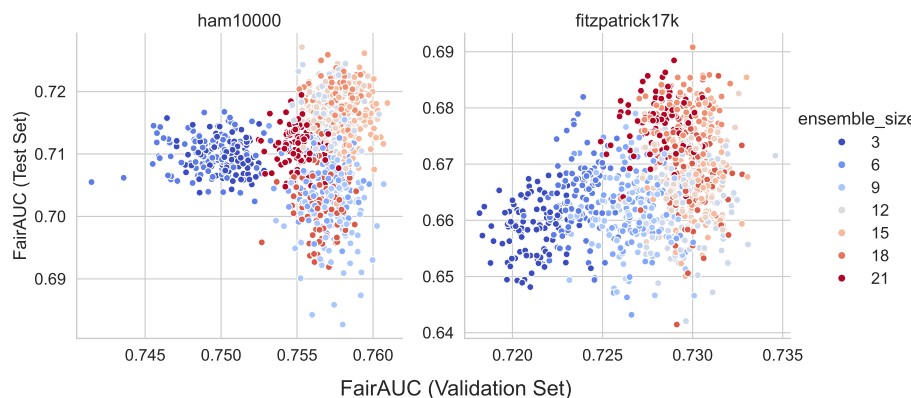

Figure 5: Relationship between FairAUC on validation (X-axis) and test set (Y-axis) across ensemble sizes. The relationship is too noisy to guide model selection.

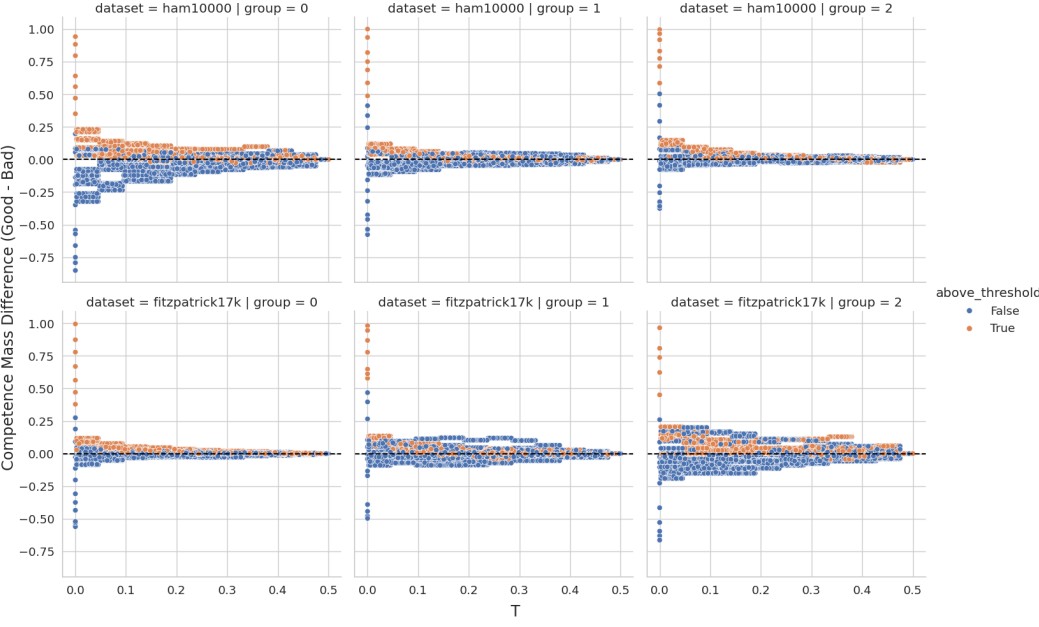

Figure 6: **Empirical validation of competence proofs**. We show that enforcing minimum recall, $k > 0.5 + \delta$, leads to *competent* ensembles (see section 3.2). $\delta$ depends on the data size (section 3.2.3) and $0.5$ comes from our proof in section 3.2.2.

Table 6: Single-image inference

| | Latency (ms) ↓ | |
| Method | CPU | CUDA |
| --- | --- | --- |
| ERM | $112.22 \pm 13.58$ | $5.42 \pm 0.31$ |
| Ensemble | $107.15 \pm 12.41$ | $5.83 \pm 0.38$ |

# E  EMPIRICAL VALIDATION OF COMPETENCE

We empirically validate our proofs from section 3.2.3 and section 3.2.2. Specifically, we want to show that enforcing recall at $k > 0.5 + \delta$ leads to competent ensembles if $\delta$ is matches the size of the datasets. This would help validate both theoretical extensions of Theisen et al. (2023).

To conduct this analysis, we set threshold $= k + \delta = 0.7$ (as described in Appendix A). We then run the competence calculations from Theisen et al. (2023) for different $k$ above and below the threshold. The resulting figure is Figure 6.

# F  BENCHMARKING EFFICIENCY

A big advantage of our FAIRENSEMBLE method is that it is efficient for training and inference because it utilises a shared backbone. In this section, we provide evidence for these claims.

The results for inference can be seen in Table 6. Here, we see comparable inference speeds for ERM and ensemble across both CPU and GPU. The GPU runs are done on an NVIDIA H100 80GB GPU. The runs are with a batch size of 1, averaged over 100 runs, with a warm-up size of 10. There are no significant differences between the methods.

The results for training can be seen in Table 7 based on Weights & Biases data (Biewald, 2020). Here, we see a larger difference; ensembles take approximately 3x longer to train compared to ERM. This may be because we are in essence training 84 times more classifiers (21 members with four heads each). Still, because of the small size of the datasets, the training times are manageable.

It is worth noting that substantial optimisation is available for training. Because the backbone is frozen, the entire evaluation set (validation sets + test set) can be pre-computed. This would drastically speed up the training. However, these optimisations were not done in the interest of time.

Table 7: Average training runtime (in minutes)

| Training Method | Avg. Runtime (min) | Std. Dev. (min) |
| --- | --- | --- |
| Ensemble | 31.79 | 5.13 |
| ERM | 8.51 | 2.28 |

# G  DERIVATIONS

## G.1  MINIMUM VALIDATION AND EVALUATION SIZES

**Statistical Framework**  We can frame the problem of ensuring minimum recall as a one-sided hypothesis test:

$$H_0 : p_{\text{val}} = p_{\text{test}} = k \quad \text{vs.} \quad H_A : p_{\text{val}} > k. \tag{10}$$

Where $p_{\text{val}}$ is our threshold of interest. Because both the test set and validation sets are small, they both introduce sampling variability. Thus, we will explicitly account for the size of both.

The hypothesis-testing framework has a few assumptions. First, it assumes that the validation and test sets are *independently* drawn from the same distribution (an assumption we explicitly follow; see

section 4.1). Second, it assumes that each positive instance is an independent **Bernoulli trial** that is either a true positive or a false negative. Finally, it assumes an approximately normal distribution. The normality assumption is met by the *Large Counts Condition*, which heuristically states that $\min(mk, m(1-k), nk, n(1-k)) \geq 10$, which in our case simplifies to $\min(\frac{m}{2}, \frac{n}{2}) \geq 10$. We thus need roughly **20** positive instances of any group in both test and validation as a minimum.

**Deriving minimums** Under $H_0$, the standard error of the difference between the minimum recall proportions in the validation and test set is:

$$\mathrm{SE}_0 = \sqrt{k(1-k)\left(\frac{1}{m} + \frac{1}{n}\right)}.$$

The one-sided $z$ statistic fixing $p_{test} = k$ is

$$z = \frac{p_{\mathrm{val}} - k}{\mathrm{SE}_0}.$$

Requiring a significance level of $\alpha$ (i.e., $z \geq z_{1-\alpha}$) yields the minimal observable validation recall:

$$p_{\min} = k + z_{1-\alpha}\sqrt{k(1-k)\left(\frac{1}{m} + \frac{1}{n}\right)}.$$

For $k = 0.5$, this simplifies to the result in Eq. 5.

## G.2 DERIVATION OF EQUAL OPPORTUNITY BOUNDS

We derive the fairness bounds for ensembles under approximate equal opportunity (or accuracy) constraints.

Starting from the definition of $k'$-approximate fairness for the ensemble, we have

$$k' = \max_{g \in \mathcal{G}} \mathbb{E}_{h \sim \rho}[L_g(h)](1 - \mathrm{EIR}_{g^*}) - \min_{g \in \mathcal{G}} \mathbb{E}_{h \sim \rho}[L_g(h)](1 - \mathrm{EIR}_{g^*}) \tag{11}$$

$$\leq \max_{g \in \mathcal{G}} \mathbb{E}_{h \sim \rho}[L_g(h)] - \min_{g \in \mathcal{G}} \mathbb{E}_{h \sim \rho}[L_g(h)](1 - \mathrm{EIR}_{g^*}) \tag{12}$$

$$\leq k - \min_{g \in \mathcal{G}} \mathbb{E}_{h \sim \rho}[L_g(h)] \cdot (-\mathrm{EIR})_{g^*} \tag{13}$$

$$\leq k + \max_{g \in \mathcal{G}} \mathbb{E}_{h \sim \rho}[L_g(h)]\mathrm{DER}_{g^*} \tag{14}$$

where $g^*$ is an appropriate distribution (e.g., positives, negatives or all points) constrained to a particular group $g$. By substituting in the lower bound from Theorem 2 instead of 0, we obtain the slightly tighter bound of Equation 4.

# H DETAILED RELATED WORK

**Fairness in Medical Imaging** Deep learning-based computer vision methods have become highly popular for medical imaging applications (Cai et al., 2020), yet despite achieving near-human performance on top-level metrics (Liu et al., 2020), they consistently underperform for marginalised groups (Xu et al., 2024; Koçak et al., 2024). These biases persist across different domains and modalities from dermatology (Daneshjou et al., 2022) to chest X-rays (Seyyed-Kalantari et al., 2021) and retinal imaging (Coyner et al., 2023). For instance, there is pervasive bias in skin condition classification (Oguguo et al., 2023; Daneshjou et al., 2022; Groh et al., 2021), likely due to both bias in data collection (Drukker et al., 2023) and treatment procedures (Obermeyer et al., 2019).

The sources of unfairness arise from different stages in the development process (Drukker et al., 2023). One persistent issue is unbalanced datasets (Larrazabal et al., 2020). Unbalanced datasets can lead to insufficient support for disadvantaged groups, which can lead to worse representations and more uncertain results (Ricci Lara et al., 2023; Mehta et al., 2024).

A successful approach to mitigating fairness is to do extensive hyperparameter and architecture search (Dutt et al., 2023; Dooley et al., 2022). By jointly optimising for fairness and performance, these

methods can reduce the generalisation gap and outperform other methods. However, because of their computational cost, we do not compare against these in this work. However, our method can be built on top of the backbones found by the architecture search.

Defining fairness in the context of medical imaging is another challenge. While traditional fairness metrics, like equal opportunity (Hardt et al., 2016), are concerned with minimising disparities between groups, this might not be appropriate in a medical context. For instance, Zhang et al. (Zhang et al., 2022) find that methods which optimise this notion of group performance reduces the performance of all groups. This phenomenon of 'levelling down' (Zietlow et al., 2022) can have fatal consequences for patients and not meet the legal standards of fairness (Mittelstadt et al., 2024). Instead, researchers should strive to enforce minimum rate constraints, i.e., the performance of the worst-performing groups, which can help reduce persistent problems of underdiagnosis and undertreatment of disadvantaged groups (Seyyed-Kalantari et al., 2021).

**Fairness in Hate Speech**   A key issue in hate speech detection is multilingual disparities (Tonneau et al., 2025). Hate speech detection models and datasets are predominantly build for an American English context (Tonneau et al., 2024). Blindly trusting that detection models scale across languages and contexts can lead to catastrophe such as with the anti-muslim violence in Myanmar (Deejay et al., 2024).

Since low-resource languages, by definition, lack data, existing fairness methods fall short for the same reasons as in other domains. Existing methods either work on large(r) datasets (Gupta et al., 2025) or lack a joint evaluation of fairness and performance (e.g., Bauer et al., 2025). Through our analysis, we demonstrate that ensembles can enhance fairness—even in low-data scenarios.

In terms of appropriate fairness metrics, there is a more direct trade-off between false positives (which hurt the falsely accused offenders) and false negatives (which hurt the victims). Which to prioritise depends on the specific context of the application. Still, similar risks of 'levelling down' are present (Mittelstadt et al., 2024).

