# OpenReview forum: "Fairness by Design: Efficient Fair Ensembles for Low-Data Classification"
_ICLR.cc/2026/Conference — ICLR 2026 Conference Withdrawn Submission_

### Official Review · Reviewer_p8PR · 2025-10-22

**Soundness:** 2
**Presentation:** 2
**Contribution:** 2
**Rating:** 2
**Confidence:** 3

**Summary:**

The submission dicusses the problem of guaranteing the fairness for ensemble classifiers, if individual classifiers are known to be fair. Specifically, it extends a result from (Theisen'23) about the accuracy of ensembles to notions of fairness that can be expressed as the accuracy on a subset of the data (or a restricted data distribution), such as controlling the false-negative rate (=recall) by measuring the accuracy on the set of negative examples (=recall). The main results is that fairness of the ensemble is at least as high as that of the participating classifiers as long as these are independent and have at least 50% recall (analogously to the 50% individual accuracy criterion of (Theisen'23). The submission also states how many samples are needed to ensure this reliably from finite data under approximate Gaussianity.
While the contribution is mostly conceptual, the manuscript also reports experiments, indicating that the promised fairness properties hold on real-world datasets.

**Strengths:**

+ the topic of ensemble fairness is relevant and interesting.
+ the described analysis appears correct and provides useful answers in some setting.
+ the viewpoint of treating recall as accuracy on a subset might be useful for other problems as well.
+ details supplemental material and high quality source code is provided.

**Weaknesses:**

The manuscript has a number of shortcomings, mostly in terms of presentation,
and a limited scientific contributions.

* Framing

The manuscript argues that ensembles are relevant in the small-data regime, but it does not provide any evidence for this. The title states "fairness by design", but the contribution really not about any design, but the fairness is simply enforced by increasing each classifier's recall threshold. The title also claims "efficient" ensembles, but efficiency is not explicitly studied or modelled, it only emerges from the use of a deep architecture with pretrained shared backbone (which is independent of any fairness aspect or theoretical guarantees).

* Clarity of presentation

I found the presentation to be unclear in some parts.
The work crucially relies on (Theisen'23), from which is adopts the definition of a *compenent* ensemble and the guarantee that ensembling cannot make the accuracy worse. However, the way the prior work is presented makes it hard to understand the prior work. For example, $C_{\rho}$ is taken from Assumption 1 in (Theissen'23). However, this quantity depends on the threshold $t$. In the submission, $t$ is not mentioned or quantified, leaving the definition incomplete. I had to consult (Theissen'23) to see that the condition should hold for all possible $t\in[0,0.5]$. The main result (Theorem 1) of (Theissen'23) is that ensembling never makes accuracy worse. The submission consistently phrases this fact as that it *improves* performance (I would suggest to avoid 'performance' anyway, as it can be mistaken for a property of runtime, not quality).
The definition of *restricted ensemble error* (line 186) includes an expectation over the data distribution, while the prior definition of error rate was a per-datapoint quality. I assume this is a mistake, as otherwise it would be a deterministic quantity and the following probability statements (line 176) would not make sense.
A central observation in the submission is "Minimum recall corresponds to competence on g+, minimum sensitivity to competence on g−, overall accuracy to competence on the full dataset" (line 192).
This seems wrong, or at least it is unclear to me. The *restricted ensemble error* should correspond to 1 minus recall, as it specifies the probability of a false negative (positive label, negative decision).
*Competence* is defined as the difference between the probability of two error rate intervals, I do not see the relation to recall here. Again, the threshold $t$ is not discussed here.

I also found the presentation to be rather imbalanced. The manuscript, e.g., presents a proof sketch of a special case of a Jury theorem (line 264ff), which could be skipped or moved to the appendix, but other parts of too brief to allow, e.g., the exact data split procedure for training the ensembles is missing, which is important to understand e.g. if the assumption of independence between ensemble members is fulfilled.
The description how fairness is enforced (line 123ff) is quite brief. It appears to use not the ground truth protected attribute but its learned prediction as a source, which is a clear source of bias that could invalidate the guarantees. It seems that the procedure can only enforce overall per-classifier recall, but definition of competence requires a positivity conditions between all intervals $t, 1/2)$ and $[1/2, 1−t]$.


* Scientific contribution

I find the scientific contribution quite limited. Apart from questions about correctness (see above),
the main observation is that (Theissen'23)'s results can be transferred from overall accuracy to accuracy of suitable subsets, which recovers, e.g., recall. The subsequent mathematical analysis is rather straight-forward, including the derivations of a necessary sample size (5) based on approximate normality. A further shortcoming is that the results require independent errors between ensemble members (which are not fulfilled for standard bootstrap/cross-validation samples in practice), and that the condition of at least 50% recall emerges, which limits the tasks where this result is applicable.

* Experimental evaluation

The experiments appear valid (with the caveat that they are not reproducible from the manuscript alone, one has read the code). It is not clear, though, how exactly they relate to the main scientific contribution. The statement of the manuscript is that fairness is at least preserved from ensemble members to the majority vote classifier. This is a new form of analysis, but not a new *method* that could be shown to be superior to other methods. Comparing the resulting fairness to other fairness methods or ensembles on non-fair classifiers is good to see, but tangential to the main statement.
Furthermore, one important baseline is missing: given that for the base classifiers the fairness is forced by adjusting decision thresholds on a validation set, what stops the user from doing so for the resulting ensemble classifier? The latter would be required baseline. In the studied setting, is it better at all to train an ensemble of fair classifiers, or would it be enough to train a standard ensemble and make it fair afterwards?

**Questions:**

* please address my concerns about the theoretical description and the mapping to the experimental setup.
* please discuss my question about the baseline of making the majority-vote classifier fair post-hoc (you do not have to run such an experiment, as I do not believe that author responses should contain new experiments).

In case my low scores were based on a misunderstanding of your contribution, I will be happy to adjust them.

---

> ### Author Response · Authors · 2025-11-19
> **Clarifying theory, contribution of new method, and evaluation.**
>
> Thank you for your helpful comments, particularly regarding the clarity of presentation of the theory. We are glad you acknowledge our high-quality evaluation and source code.
>
> ## Framing & Contribution
> > Framing: The manuscript argues that ensembles are relevant in the small-data regime, but it does not provide any evidence for this. The title states "fairness by design", but the contribution is really not about any design [...]
>
> "By design" means performed with intent. We explicitly enforce fairness (including minimum recall) on ensemble members for very small group data (Table 1) using Delaney et al's method, not by increasing per-classifier thresholds. See section 3.1.
>
> > The title also claims "efficient" ensembles, but efficiency is not explicitly studied or modelled [...].
>
> **Action 1: We will more clearly signpost our efficiency comparisons.**
>
> We empirically study efficiency gains in Appendix F (signposted L150), comparing inference latency (Table 6) and training run-time (Table 7), showing small overheads. Since FairEnsemble requires auxiliary heads for per-member OxonFair, efficient stratified fold division is core to our method's effectiveness.
>
> > I find the scientific contribution quite limited. [...] the main observation is that (Theissen'23)'s results can be transferred [...].
>
> Our main contribution is a novel ensembling method that enforces fairness in challenging low-data scenarios. The theoretical guarantees illustrate _why_ and _when_ our method outperforms ERM. We outperform existing benchmarks on accuracy-fairness trade-offs (section 5).
>
> > A further shortcoming is that the results require independent errors between ensemble members [...], and that the condition of at least 50% recall emerges, which limits the tasks where this result is applicable.
>
> First, the results **do not** require independent errors - they hold for mildly correlated members (L262-264). We empirically validate ensemble competence in Appendix E (signposted L316).
>
> Second, our FairEnsemble method explicitly enforces the 50% recall condition (section 3.1). Fig. 3 (left and centre) shows we can always enforce solutions satisfying this condition.
>
> ## Clarity of presentation (of theory)
> > In the submission, $t$ is not mentioned or quantified, leaving the definition incomplete.
>
> **Action 2: We will further clarify the interpretation of $t$**
>
> In Theisen'23, $t$ is a margin parameter varied in the definition (Table 5). We'll be more explicit in the main text.
>
> > The submission consistently phrases this fact [Theisen et al.] as that it improves performance (I would suggest to avoid 'performance' anyway, as it can be mistaken for a property of runtime, not quality). [...]
>
> Theisen'23 shows this for accuracy. By restricting datapoints to those with positive ground-truth (y=1), the result holds for recall; restricting to y=0 gives specificity, since accuracy over positives equals recall.
>
> By further restricting to particular groups, Theisen's result implies worst group accuracy, recall, and specificity do not degrade.
>
> We'll clarify that Theisen only referred to accuracy.
>
> > The definition of restricted ensemble error (line 186) includes an expectation over the data distribution, while the prior definition of error rate was a per-datapoint quality. [...]
>
> **Action 3: We will fix the notation error.**
>
> There's a typo in line 186: the expectation should be only on $\rho$, i.e., $\mathbb{E}_{h\sim\rho}$. Dataset dependence $(X,Y)\sim\mathcal{D}$ should only be for $P$ in Eq. 1.
>
> >  A central observation in the submission is "Minimum recall corresponds to competence on g+, minimum sensitivity to competence on g−, overall accuracy to competence on the full dataset" (line 192). This seems wrong [...]
>
> - **Action 4: We will tighten language around "correspond".**
> - **Action 5: We will clarify the distinction between "competent" and "competence".**
>
> An ensemble being competent on g+ provides guarantees for minimum recall; g- provides minimum precision guarantees.
>
> We acknowledge the confusing distinction between "competence" (L175 quantity) and being "competent" (Theisen'23 condition). Our claims regard what competent ensembles imply for fairness and enforcement.
>
> ## Experimental Evaluation
>
> > It is not clear how [the experiments] relate to the main scientific contribution.
>
> As mentioned above, our main scientific contribution is our proposed method, FairEnsemble, which we show to improve on baselines for very challenging datasets. Thus, the method (which efficiently uses OxonFair enforced heads) is new and warrants comparisons.
>
> **Thank you again for taking the time to review the paper and providing helpful feedback! Do the above actions address your concerns with the paper? If not, what further clarification or modifications could we make to improve your score?**

---

> > ### Comment · Reviewer_p8PR · 2025-11-23
> > **Thank you for the response**
> >
> > Dear authors,
> >
> > thank you for providing your response. However, I still believe the submitted manuscript has too many shortcomings (which I highlighted in my review) for publication at a top venue for ML research such as ICLR. Your answers provide context and interpretation for how you see your own contribution, but they do not solve the actual issues. Therefore, I keep my recommendation.
> >
> > -- Reviewer p8PR

---

### Official Review · Reviewer_SM5J · 2025-10-26

**Soundness:** 2
**Presentation:** 3
**Contribution:** 2
**Rating:** 4
**Confidence:** 4

**Summary:**

This paper tackles the problem of algorithmic fairness in low-data and imbalanced regimes, which are common in domains such as medical imaging and hate speech detection. The authors propose **FAIRENSEMBLE**, an ensemble framework of fair classifiers that shares a pretrained backbone while enforcing fairness constraints at the member level. During validation, each ensemble member optimizes a weighted combination of accuracy and fairness (minimum recall or equal opportunity), and final predictions are aggregated by majority voting.
Theoretically, the paper adapts _ensemble competence_ theory (Theisen et al., 2023) to group fairness, showing that (i) enforcing minimum recall above 0.5 guarantees ensemble competence for positive samples, and (ii) fairness constraints such as error parity remain bounded across groups. Empirically, FAIRENSEMBLE outperforms baseline fairness methods (OxonFair, FairRet, ERM) on three medical imaging datasets (HAM10000, Fitzpatrick17K, FairVLMED) and one hate speech dataset, achieving better fairness–accuracy trade-offs measured by the proposed **FairAUC** metric.

**Strengths:**

+ Addresses a **high-impact but underexplored problem**: ensuring fairness under small and unbalanced datasets, a realistic setting for medical AI.
+ Presents a **clear and coherent methodology**, integrating fairness constraints into ensemble training with shared backbones to improve data efficiency.
+ Provides **formal theoretical analysis** connecting ensemble competence and fairness guarantees, grounding the method in provable conditions.
+ Introduces **FairAUC**, a simple yet effective metric to evaluate the fairness–accuracy Pareto frontier.
+ Demonstrates consistent empirical improvements across both imaging and NLP tasks, with notable gains in fairness metrics over strong baselines such as OxonFair.

**Weaknesses:**

+ **Limited theoretical originality**: the core proofs and guarantees rely heavily on prior results from Theisen et al. (2023) and Condorcet Jury Theorem. The paper adapts these to group fairness rather than providing novel derivations.
+ **Fairness validation lacks per-group evidence**: experiments report global metrics (FairAUC, DEO) but do not show _which_ disadvantaged groups actually improve (e.g., recall on minority skin tones). The fairness claim is therefore not fully substantiated.
+ **Fairness metrics and theoretical assumptions are not perfectly aligned:**
the theoretical guarantee relies on each ensemble member (or fold) achieving a minimum recall greater than 0.5, but the experiments do not verify whether this condition actually holds for all folds. As a result, it remains unclear whether the theoretical fairness guarantees are truly satisfied in practice.

**Questions:**

1. Can the authors provide **per-group confusion matrices or recall statistics** to verify that the observed improvements indeed arise from better recall of disadvantaged groups, rather than overall averaging effects?
2.  The theoretical guarantee relies on minimum recall ≥ 0.5. How often does this condition hold in practice across datasets?
3. The proposed FairAUC metric is an interesting way to quantify fairness–accuracy trade-offs. Could the authors clarify how well it correlates with standard fairness metrics (e.g., Equal Opportunity, Equalized Odds) and whether it can consistently reflect group-level improvements?

---

> ### Author Response · Authors · 2025-11-19
> **Clarification of Contribution, Evaluation, and Theory**
>
> Thank you for your helpful comments, particularly regarding how we evaluate fairness and clarifying our contribution. We are glad you acknowledge our "clear and coherent methodology" and how our FairAUC provides a "simple, yet effective" way to measure fairness-accuracy trade-offs.
>
> ##  Contribution
> > Limited theoretical originality [...]
>
> We would like to underscore that our primary contribution is our new method, FairEnsemble, which outperforms existing methods in very low-data scenarios. The theoretical results primarily substantiate _when_ and _why_ our method works.
>
> While we appreciate that we rely on established results to show why fairness works, we do believe that the application to fairness provides useful clarity. Our position is simple: if something has not been done before, then it is novel - regardless of whether it adds complexity. Papers that propose simple modifications that are obviously useful will rarely be accepted if novelty is seen as complexity. However, such papers are often more useful both for practitioners (it’s easy to change one component) and scientifically (a single modification makes it easy to see what actually makes a difference).
>
> Furthermore, another key contribution of our paper is the FairEnsemble method that makes it possible to design fair ensembles - instead of just relying on the inherent fairness properties of ensembles as discussed by previous works.
>
> ## Fairness evaluation
> > Fairness validation lacks per-group evidence [...]
>
> > Can the authors provide per-group confusion matrices or recall statistics to verify that the observed improvements indeed arise from better recall of disadvantaged groups, rather than overall averaging effects?
>
> **Action 1: We will add _which_ group exhibits improvement in results.**
>
>
> Definitions of group fairness concern the relative performance of different groups, either through disparity measures (such as DEO) or minimum rates (such as minimum recall). As such, FairAUC does substantiate fairness claims by showing how the worst-group performance changes - it is not a global metric, but a way to marginalise over thresholds (see Eq. 6). As discussed in L107-112, parity measures can "level-down", which is why we predominately report minimum rate aggregates (like FairAUC).
>
> That said, we will specify _which_ group performs worst for increased clarity.
>
> > Could the authors clarify how well [FairAUC] correlates with standard fairness metrics [...] and whether it can consistently reflect group-level improvements?
>
> Unlike standard measures such as EOp and Eodds, FairAUC requires minimum recall rates to be satisfied (i.e., the recall rate for every group is above k% for a range of values k%). Minimum rates are never improved by levelling down (or decreasing recall or sensitivity for some groups), but only by improving the lowest group rates.
>
> FairAUC is a way to marginalise over different minimal rates. Where the fairness method doesn't naturally support minimal rates, we simply vary a global threshold until particular minimal rates are obtained. By definition (see Eq. 6, L362), FairAUC reflects the performance of the worst-performing group consistently.
>
> There's no guarantee that minimum recall rate and EOp will correlate, but in practice directly optimizing for minimum recall rate will eventually result in EOp of zero (see Mittlestadt et al. '23) Similarly, going the other way and enforcing EOp, and then considering a sweep of global thresholds, resulting in different (minimum recall) rates is a reasonable heuristic reflecting how fair algorithms may have to be used in practice.
>
>
> ## Theory
> > Fairness metrics and theoretical assumptions are not perfectly aligned [...]
>
> > The theoretical guarantee relies on minimum recall ≥ 0.5. How often does this condition hold in practice across datasets?
>
> **Action 2: We will further clarify the assumptions.**
>
> As mentioned in L315-316, we do empirically validate that our ensembles have _competence_ (which is the main theoretical assumptions vis a vis Theisen'23). To clarify the assumptions of our proofs: First, we only assume that the minimum recall exceeds 0.5 on average due to generalisations of the theorem (see L262 for references).
>
> Furthermore, a core contribution of our FairEnsemble technique is that we can enforce arbitrary minimum recall thresholds for the individual members. This means that we can guarantee fairness (in contrast to previous observational works like Claucich et al., 2025). While there will inevitably be statistical drift between the validation set (where we enforce fairness) and the test set, we set statistical bounds in section 3.2.3.
>
>
> **Thank you again for taking the time to review the paper and for your helpful feedback! Do the above actions address your concerns with the paper? If not, what further clarification or modifications could we make to improve your score?**

---

### Official Review · Reviewer_n89Z · 2025-10-28

**Soundness:** 2
**Presentation:** 1
**Contribution:** 2
**Rating:** 4
**Confidence:** 4

**Summary:**

The paper proposes an efficient deep-ensemble framework that enforces group-fairness constraints at the member level and argues these constraints are preserved at the ensemble level. The method shares a frozen ImageNet-pretrained backbone across multiple classifier heads, where each member is trained on a stratified fold and equipped with auxiliary 'protected-attribute' heads. At validation time, the method tunes head weights to enforce either minimum recall or equal opportunity, while maximizing accuracy. Finally, the predictions are aggregated by majority vote at inference.

Theoretical results show that (i) if each member meets a minimum rate and the group is competent, the ensemble also meets that minimum rate; (ii) for error-parity metrics, approximate parity among members leads to bounded disparity in the ensemble; and (iii) under independent errors, ensuring recall > 0.5 for each member guarantees ensemble-level recall > 0.5. Empirically, the method improves fairness-accuracy balances on CV tasks and hate-speech detection, measured with FairAUC.

**Strengths:**

* Focus on low-data fairness where parity metrics can level down performance; the choice of minimum recall as a primary fairness target for safety-critical medical tasks is appropriate and clearly argued.

* A shared (frozen) EfficientNetV2 backbone with masked multi-heads makes M-member ensembling cheap; inference is near-single-model cost while retaining diversity from fold-specific heads. Clear training/validation partitioning and majority voting.

* Extends ensemble competence to group-restricted subsets and links guarantees to minimum-rate constraints; provides (approximate) error-parity bounds and a validation/evaluation sample-size formula intended to guide split sizes.

* Authors provide anonymized code, deterministic seeds, and explicit compute details.

**Weaknesses:**

- *Presentation*. As a note, the paper names their method FAIRENSEMBLE. In my opinion, this causes confussion with the literature, since one of the already identified close works is Ko et al "Fair-Ensemble: When Fairnes...", which they (Ko et al) use in the title and in the section 3 to name the effect of some ensembles improving the workst group accuracy in certain tasks. I suggest renaming the method to avoid confusion.


- *Contribution and Related Work*. Authors claim that related work does not do theoretical analysis on the fairness of ensembles (l.43, l.156, l.287). However this is not totally true. For instance Grgić-Hlača et al focus on theoretical considerations on ensemble fairness, specifically about conditions inder when ensembles are guaranteed to be fair or not. In additon, Sweighofer et al. also provide theoretical insights on the potential causes of the fairness issues in ensembles, and propose a bayesian perspective on the average predictive diversity.

- *Related work*. The sentence in the intro "Existing fairness interventions often fail in these low-data settings" (l.37) must be supported by citations or evidence. The current citation Zong et al does not support this claim.

- *Related work*. The paper (l.97) currently states that ensembles sometimes help fairness and sometimes don't, but it doesn't disentangle when and why. Please be precise about: (i) ensemble heterogeneity (heterogeneous vs. homogeneous members), (ii) task modalities (tabular vs. vision vs. language), and (iii) fairness metrics (e.g., minimum per-group accuracy/recall, disparate impact, equalized odds/opportunity). A structured and more extended related-work paragraph or even table separating these axes would let readers see under which conditions prior work finds gains or not.

- *Related Work*. I also miss discussion with more works in the fairness in deep ensembles. Please I recommend the authors to look in the Sweighofer et al., and Claucich et al. papers, which are very related to the topic of this paper. Examples of relevant missing literature are Gohar et al. (ensembles of shallow models), Grgic-Hlaca et al. (theoretical considerations on ensemble fairness) or Bhaskaruni et al. (Fair AdaBoost).

- *Contribution*. Several guarantees rely on independence correlation of member errors. With a shared frozen backbone, heads may be substantially correlateds. The paper cites disagreement/DER conceptually, but I could not find per-group empirical DER/competence measurements on validation and test (especially on the positive groups $g^+$). Adding these would strengthen the claims.

- *Contribution*. The main contributions of the authors build upon known ensemble-competence results (Theisen et al. 2023, Condorcet variants) to subgroup-restricted sets and fairness metrics. This is valuable, but the mathematical steps feel incremental. Clarifying what is genuinely new (beyond restatement with restricted distributions) would help to highlight the contributions.

- *Contributions and Evaluation*. Although authors propose their theoretical framework for EOp and minimum-rate constraints, the main focus of their empirical evaluation focuses on FairAUC, which limits the interpretability of their contributions. EOp and min recall are just mentioned in Fig 3 center and right, but no discussion about the theoretical insights and empirical findings have been done on min recall and EOP. Including discussion about this connection would strengthen both the empirical and the theoretical contribution of the paper.

- *Evaluation*. To enable clearer comparisons with the already identified close work (e.g., Schweighofer et al.; Ko et al.; Claucich et al.), include standard public benchmarks commonly used in this literature for computer vision areCelebA, LFWA, CheXpert, FairFace. However, the paper would be benefited from including tabular benchmarks as well (COMPAS, Adult, German Credit, ACSIncome...). This also helps stress-test across modalities and metrics (parity and minimum-rate).

- *Evaluation*. Given how closely the paper builds on Schweighofer et al., Ko et al., and Claucich et al., these must be baselines. Schweighofer et al. propose an ensemble threshold post-processing for fair decisions, leveraging better calibration for group-aware threshold adjustment. Ko et al. analyze initialization and batch order as primary sources of diversity in deep ensembles. Finally, Claucich et al. show that equal rebalancing can be harmful and that over-representing the minority when their task is harder can mitigate harms, which is of critical importance for author's focus on scenarios where data is scarce and unbalanced across groups. These are relatively simple, pre or post-processing techniques. Without these, it's hard to attribute the authors contributions's gains vs. known ensemble/thresholding/calibration/diversity/sampling effects. In addition, there are more relevant baselines in the literature that should be included (e.g., Bhaskaruni et al.), missing due to the lack of depth in the literature review.

- *Evaluation*. I'd like to see ablations on the backbone. For instance, frozen vs. fine-tuned backbone, and without protected-attribute heads (how much does the surgery vs. ensembling contribute?) would help isolate which component drives the gains.

- *Presentation*. Add consistent venue/URL/DOI across all references, especially the closest prior work.
  - Add conference (or arxiv) in all references (especially in the ones that the paper closely relies on (!!)), for instance:
    * !! Schweighofer et al. is in ICML 2025 https://proceedings.mlr.press/v267/schweighofer25a.html
    * !! Claucich et al. is in FAccT 2025 https://dl.acm.org/doi/10.1145/3715275.3732200
    * !! Ko et al. is in arXiv only, but add it https://arxiv.org/abs/2303.00586
    * Chen et al. is in arXiv only, but add it https://arxiv.org/abs/2007.00649
    * Jiménez-Sánchez et al. is in FAccT 2025 https://dl.acm.org/doi/10.1145/3715275.3732035
  - Remove or footnote non-academic sources (e.g., blogposts) to avoid confusion (e.g. Biewald).

---
*Refs*
* Gohar et al. Towards understanding fairness and its composition in ensemble machine learning
* Grgić-Hlača et al. On fairness, diversity and randomness in algorithmic decision making
* Bhaskaruni et al. Improving prediction fairness via model ensemble.

**Questions:**

In addition to the weaknesses mentioned above, I have some minor questions for the authors that I would like clarified.

* You state the approach 'applies equally to other group fairness metrics.' Could you demonstrate at least one additional metric (e.g., minimum precision or FNR parity) to corroborate the theory claim

* How exactly are head weights tuned on validation to meet fairness constraints? Is this a constrained optimization, a grid search...? What are the runtime/feasibility behaviors when constraints are tight?

* Did you try lightly fine-tuning the backbone to improve diversity vs. overfitting? Any change in DER, fairness?

---

> ### Author Response · Authors · 2025-11-19
> **Clarifying related works, baselines, and contribution**
>
> Thank you for your helpful comments, particularly regarding positioning our paper in related works and clarifying our contribution. We are glad you acknowledge the importance of developing methods for minimum rate fairness in medical imaging and the efficiency of FairEnsemble.
>
> ## Presentation
> > The paper names their method FAIRENSEMBLE. [...]. renaming the method to avoid confusion.
>
> FAIRENSEMBLE was an anonymous placeholder. We will update the name.
>
> ## Related Work
> > Authors claim that related work does not do theoretical analysis on the fairness of ensembles [...]
>
> **Action 2: We will clarify novelty vis-à-vis these papers in related works and theory sections**
>
> While we will soften our claims, we provide new theoretical insights: **interventional** results (supported by our method) that apply to **minimum rate** constraints. Grgic-Hlaca et al. focus on random members from ensembles and different fairness metrics. Nevertheless, these works are important for positioning our contributions.
>
> > The sentence in the intro "Existing fairness interventions often fail in these low-data settings" (l.37) must be supported by citations [...]
>
> **Action 3: We add citations**
>
> Zong et al. explicitly state "state-of-the-art methods do not outperform the ERM with statistical significance." noting small data with <100 data points for some groups (see our Table 1). We will add further references.
>
> > discussion with ... Sweighofer et al., and Claucich et al., Gohar et al., .. or Bhaskaruni et al..
>
> **Action 4: We will expand related works with suggestions.**
>
> We cut related works due to page limits. While familiar with Schweighofer et al. and Claucich et al., we appreciate the additional recommendations. Gohar et al. and Bhaskurani et al. both concern shallow models on tabular data. We will these to related works.
>
> ## Contribution
> > Several guarantees rely on independence correlation [...]
>
> First, the theoretical results **do not** assume independence since we rely on established jury theorems; we only provide a sketch for completeness (see L262-265). Second, we **provide** empirical competence measurements on the test set in Appendix E (signposted in L315-316).
>
> > The main contributions of the authors build upon known ensemble-competence results [...]
>
> While our _theoretical_ contributions depend on Theisen et al., our core contribution is a novel _empirical_ method that improves over baselines on low-data benchmarks. Our theory explains _why_ and _when_ it works. Deriving these guarantees for low-data deep ensembles is new. We will clarify in the main text.
>
> > Although authors propose their theoretical framework for EOp and minimum-rate constraints, the main focus of their empirical evaluation focuses on FairAUC [...]
>
> **Action 5: We will clarify the relation between FairAUC and minimum-rate constraints.**
>
> As described in L359, FairAUC marginalises over decision thresholds for minimum rate constraints. We directly compare against minimum recall and EOp in Table 2 (fairness violations) and of EOp in Table 3 (contra Delaney et al., 2024).
>
> > Given how closely the paper builds on Schweighofer et al. [etc.] these must be baselines. [...]
>
> - **Action 6: We will clarify how our existing baselines relate to existing literature**
> - **Action 7: We will add Schweighofer et al.'s baselines**
>
> These works do not apply well to low-data classification. For Ko et al., their approach wouldn't provide diversity in our convex setting.
>
> For Claucich et al., we already implement rebalancing for all methods. Selecting an "optimal" training (like they do) is infeasible--they use held-out data inaccessible to us due to data paucity. We will clarify this.
>
> For Schweighofer et al., we **implement** a similar baseline ("Ensemble"), which we outperform. We will add a closer implementation with relevant techniques (HPP etc.).
>
> > Evaluation. I'd like to see ablations on the backbone. [...]
>
> Our "ERM Ensemble" baseline excludes the protected-attribute heads, which we outperform (see fig. 2). A fine-tuned backbone would introduce more dependence between members. Since we apply the same frozen fine-tuning across baselines, this doesn't advantage our method.
>
> > Presentation. Add consistent venue/URL/DOI across all references, especially the closest prior work.
>
> We have updated the bib file.
>
> ## Questions
> > Could you demonstrate at least one additional metric [...]?
>
> **Action 8: We add FNR experiment.**
>
> In our problems, most samples are negative; high minimum precision and accuracy can be achieved by a constant classifier. Instead, we will show optimization of balanced accuracy alongside minimum FNR.
>
> > How are weights tuned [...]?
>
> We use Delaney et al., 2024 (L123-137) to find a Pareto frontier with grid search + local search. No additional computational overhead for tight constraints.
>
> **Thank you again for your helpful feedback! Do the above actions address your concerns? If not, what further modifications could improve your score?**

---

> > ### Comment · Reviewer_n89Z · 2025-11-24
> >
> > Thank you for the detailed rebuttal and for engaging carefully with the review. I still have some concerns unresolved, but first I have an **important note about the changes and updated PDF.** The authors did not upload a revised PDF during the rebuttal period, so at this stage I can only evaluate the promised changes, not their actual implementation. If the authors implement the changes they described in the rebuttal and clearly mark them in the revised manuscript (e.g., with colored text or a tracked-changes style), I could better evaluate the improvements of the manuscript. Below, I detail what I consider to be addressed, **conditional on proper implementation**.
> >
> > * **Name / presentation**: Renaming the method and cleaning up the references (venues/URLs/DOIs, and de-emphasising non-academic sources) fully addresses my presentation concerns here.
> > * **Related work & claims of novelty**: Softening the "no theoretical analysis" claim, explicitly discussing Grgić-Hlača, Schweighofer, Claucich, Gohar, Bhaskaruni, and clarifying what is genuinely new (interventional guarantees for minimum-rate constraints in low-data deep ensembles) goes a long way toward fixing the positioning issues. *Assuming this is clearly reflected in the intro, related work, and theory sections*, I consider this concern largely addressed.
> > * **Low-data fairness claim**: If you explicitly connect the statement "Existing fairness interventions often fail in these low-data settings" to Zong et al.'s small-sample results and add at least one more supporting citation, that resolves my request for evidence.
> > * **Theory / independence & competence**: Clarifying that the main theoretical guarantees do not rely on independence, and making the competence/DER-style measurements in Appendix E more visible from the main text (and explicit about group-restricted subsets) addresses this partially. I'd still encourage you to highlight, in the main text, the exact assumptions under which each guarantee holds and to clearly separate the independence-based sketch from the core results.
> > * **FairAUC vs minimum-rate / EOp**: The clarification that FairAUC marginalizes over thresholds for minimum-rate constraints, plus the existing comparisons in Tables 2-3 and additional discussion, satisfactorily addresses my concern about alignment between theory and experiments.
> > * **Additional metric**: Adding an FNR-based experiment (and explaining why precision is less informative in your imbalanced setting) directly answers my question about demonstrating another metric.
> >
> > Even assuming all the above changes are made, some **partial gaps remain significant**, and addressing them would, in my view, require **major** rather than minor changes:
> >
> > 1. **Baselines and links to closely related ensemble-fairness methods** While the authors promise to add Schweighofer-style baselines and to clarify why Ko et al. and Claucich et al. do not transfer straightforwardly, the current paper still lacks (1) A clear, implemented Schweighofer+HPP-type post-processing baseline, and (2) either (a) at least a small-scale empirical comparison to Ko/Claucich, or (b) a technically detailed argument about infeasibility or mismatch, beyond high-level justification. Given how central these works are in the "fair deep ensemble" space, better aligning baselines with them is, in my view, a major change, not a minor polishing step.
> >
> > 2. **Ablations on backbone and protected-attribute heads** The "ERM ensemble" baseline partially addresses the role of protected-attribute heads, but does not fully tease apart the (1) frozen vs. (lightly) fine-tuned backbone, and (2) with vs. without protected-attribute heads under otherwise identical conditions. A proper ablation (even in an appendix) would substantially clarify where the gains are coming from and how much is due to ensembling vs. architectural surgery.
> >
> > 3. **Benchmarks**. I understand constraints on time and space, but including a single standard CV fairness benchmark (e.g., CelebA or CheXpert) or a single tabular dataset (e.g., Adult, COMPAS) to support claims about generality and improve your limited evaluation is a valid concern.
> >
> > ---
> >
> > The promised changes significantly address several of my main concerns, especially regarding positioning, related work, and clarity of claims. However, this **depends on a revised version** that genuinely incorporates all these modifications and clearly highlights them. Given their scope (expanded related work, new baselines, additional metrics and experiments, and clarification of theoretical assumptions), these adjustments seem closer to a *major revision* rather than minor edits. Since no updated PDF is available at this stage, I am maintaining my original score, with the possibility of an increase contingent on a thoroughly revised and clearly marked manuscript.

---

### Author Response · Authors · 2025-11-19

We thank the authors for their comments.

To correct a common misapprehension. This work is not a theory paper.

We went into this project purely concerned with the question "How can fairness be enforced in practice for low data and high-dimensional imaging?"

We went into the project with 3 goals:
1. Unlike existing approaches, the method should be usable with low data -- the performance cost of holding back enough data to reliably estimate and correct error rates, often degrades the generalisation so much that it's not worth doing.
2. The method should not be prohibitively slow to run, and should have similar training and evaluation times as a standard classifier.
3. We should have some idea of the theoretical guarantees of the approach, and when we can expect it to work.

As such, the experiments address point 1. We compare with a wide range of methods (including a strong ERM baseline) and show we outperform all of them.
2. follows directly from the design of our architecture. This is a straightforward contribution but has significant practical implications.
3. follows from the application of concordant and ensemble theorems to subsets of the data. We are not looking to make a substantial contribution to concordant theory, but to show why our method is guaranteed to work.

---

### Note · Authors · 2025-11-28

I have read and agree with the venue's withdrawal policy on behalf of myself and my co-authors.